# Comparison of Synthetic Methods and Identification of Several Artificial Antigens of Deoxynivalenol

**DOI:** 10.3390/molecules28062789

**Published:** 2023-03-20

**Authors:** Li Han, Yuetao Li, Jinqing Jiang, Changzhong Liu, Jie Hou, Bo Wang, Ziliang Wang

**Affiliations:** Henan Institute of Science and Technology, College of Animal Science and Veterinary Medicine, Xinxiang 453003, China

**Keywords:** deoxynivalenol, artificial antigen, N,N′-carbonyldiimidazole

## Abstract

The purpose of this experiment was to study the design and modification of hapten molecules and artificial antigen molecules of deoxynivalenol (DON), and to compare the preparation and identification methods of four artificial antigens. According to the characteristics of the molecular structure of DON, four artificial antigen coupling methods were designed—namely, N,N′-carbonyldiimidazole (CDI), 1-ethyl-3-(3-dimethylaminopropyl)carbodiimide hydrochloride (EDC), isobutyl chloroformate (IBCF), and N-hydroxysuccinimide (NHS)—to prepare artificial antigens and detection antigens. Through ultraviolet (UV), infrared (IR), and SDS–polyacrylamide gel electrophoresis (SDS–PAGE), along with other physical and chemical identification methods and animal immunisation, the best artificial antigen coupling method was screened. The results showed that the CDI method achieved the best effect among the synthesis methods. The titre of anti-DON polyclonal antibody (pAb) produced by animal immunisation reached 1: (6.4 × 10^3^). The half inhibitory concentration (I*C*_50_) was 47.75 ng/mL, the cross-reaction rate with 3-acetyldeoxynivalenol (3-AcDON) was slightly higher at 35.3%, and there was no cross-reaction with other compounds; therefore, four artificial antigens were successfully prepared by using the molecular structure of DON. Through identification, the CDI method was screened as the best artificial antigen synthesis method, with the highest DON pAb titre, the best sensitivity, and the strongest specificity. This will lay a solid antigenic foundation for the preparation of better anti-DON monoclonal antibodies (mAbs) in the future.

## 1. Introduction

Deoxynivalenol (DON), also known as vomitoxin, is the most common B-group monofilamentous compound [1,2,3] and contaminates cereal grains consumed by humans and animals. Research shows that DON can exist in wheat, barley, corn, and rice all over the world [4,5]. The consumption of DON-contaminated feed can lead to vomiting, diarrhoea, refusal of feed, and reduced weight gain in animals. In severe cases, this causes immunotoxicity, organotoxicity, inhibition of protein synthesis, and teratogenicity, and it is also closely related to immunosuppression, Keshan disease, oesophageal cancer, and other diseases [6,7,8]. Critically, because DON is formed prior to harvesting, its occurrence cannot be eliminated in the processes of food production, resulting in ongoing contamination challenges. Young et al. [9] found that DON can still be found in pet food made from processed grains. DON has the molecular formula C_15_H_20_O_6_ and a molecular weight of 296.32. DON has one primary and two secondary hydroxyl groups, as well as an epoxide and a conjugated ketone, any of which may be related to toxicity [10]. In addition, it was listed as a class III carcinogen in the evaluation report published by the International Agency for Research on Cancer (IARC) in 1998 [11]. Therefore, DON pollution poses a great threat to human and animal health and has attracted the attention of nations all over the world [12]. In short, the frequent occurrence of DON in food and animal feed is obviously a pressing issue in food and animal husbandry [13]. The European Food Safety Agency (EFSA) has established limit standards for the maximum allowable DON levels in various foods and feeds [14,15]. At present, China stipulates a maximum DON content of 1000 ng/g in corn, wheat, and their products [16,17].

DON is one of the most important mycotoxins related to food chains, and substantial research effort has been invested in interpreting its detection mode [18]. At present, the physical and chemical methods for detecting DON pollution in food and feed offer high accuracy and sensitivity, but their cost is high, making them unsuitable for DON pollution detection in the general feed industry. In recent years, immunological detection technology has been widely used for residue analysis of small-molecule pollutants such as mycotoxins, pesticides, antibiotics, etc. [19]. The core reagent and challenge for establishing immunological detection technology is the preparation of mAbs, and the basis of high-quality core mAb reagents is the preparation of high-quality artificial antigens. Therefore, the synthesis of artificial antigens is particularly important. In conclusion, as a traditional method, immunoassays do not require special instrumentation and equipment, are suitable for onsite and high-throughput screening, and are considered to be suitable detection tools, and their development and application have progressed rapidly in recent years [20]. The purpose of this experiment was to study the design and modification of hapten molecules and artificial antigen molecules of deoxynivalenol, identify artificial antigens, and lay a solid foundation for the later preparation of anti-DON mAbs and products with higher titres, better sensitivity, and stronger specificity.

## 2. Results

### 2.1. Physical and Chemical Property Test Results of Artificial Antigens

#### 2.1.1. Determination of Protein Concentration

The protein concentrations of four artificial antigens—DON-BSA (CDI), DON-BSA (EDC), DON-BSA (IBCF), and DON-BSA (NHS)—were measured at a wavelength of 280 nm with an ultra-micro nucleic acid protein analyser as 4.6 mg/mL, 5.2 mg/mL, 5.7 mg/mL, and 6.0 mg/mL, respectively.

#### 2.1.2. UV Identification

The results are shown in Figure 1. The maximum absorption peak of bovine serum albumin (BSA) was 278 nm, and the maximum absorption peak of DON was 218 nm. The maximum absorption peaks of the artificial antigens DON-BSA (CDI), DON-BSA (EDC), DON-BSA (IBCF), and DON-BSA (NHS) prepared by the four methods were 275 nm, 276 nm, 273 nm, and 270 nm, respectively, and all of the absorption peaks were changed. The results showed that the artificial antigens prepared by the four methods were conjugated successfully.

#### 2.1.3. IR Identification

The results are shown in Figure 2. Compared with BSA, the artificial antigens DON-BSA (CDI), DON-BSA (EDC), DON-BSA (IBCF), and DON-BSA (NHS) prepared by the four methods all displayed the amino acid characteristic peaks of the carrier protein BSA. Their IR absorption spectra were similar in the regions of 2500–3200 cm^−1^ and 1500–1660 cm^−1^, indicating the presence of BSA in the artificial antigens. Compared with the infrared absorption results of the artificial antigens and DON, there was similar absorption in the region of 1050–1150 cm^−1^, which was the characteristic peak produced by the alcohol hydroxyl group in DON, while BSA showed no absorption here, indicating that the artificial antigens contained DON. The results showed that the four artificial antigens were synthesised successfully.

#### 2.1.4. SDS–PAGE Identification

The results are shown in Figure 3. The molecular weight of BSA is approximately 66.4 kilodaltons (KD), while that of DON is 0.296 KD. The protein bands of the artificial antigens DON-BSA (CDI), DON-BSA (EDC), DON-BSA (IBCF), and DON-BSA (NHS) prepared by the four methods were all higher than those of BSA, indicating that the artificial antigens moved slower than BSA—that is, the molecular weight of the artificial antigens was higher than that of BSA. It can be preliminarily proven that the four artificial antigens were synthesised successfully.

### 2.2. Immunological Characteristic Test Results of Artificial Antigens

#### 2.2.1. Potency Determination

The results are shown in Table 1 and Figure 4. Six-week-old female BALB/c mice were immunised with artificial antigens synthesised by four methods, with four groups of five mice per group. After five immunisations, the titres were determined by indirect ELISA. One mouse with the highest titre was selected from each group, and a total of four mice were selected for comparison. The titres of pAb all reached 1: (1.6 × 10^3^). The results showed that the titres of the four artificial antigens all met the requirements but, according to the evaluation of immune effects, the order from high to low was DON-BSA (CDI), DON-BSA (EDC), DON-BSA (IBCF), and DON-BSA (NHS).

#### 2.2.2. Sensitivity Determination

The results are shown in Table 2 and Figure 5. In each group, the four mice with the highest titre values exhibited inhibitory effects on DON, and all of them met the criterion of a linear relationship. Among them, the DON-BSA (CDI) group had the best inhibitory effect; the linear regression equation was y = −37.056x + 112.23, R² = 0.9744, and the I*C*_50_ was 47.75 ng/mL. The DON-BSA (EDC), DON-BSA (IBCF), and DON-BSA (NHS) groups had the second-lowest sensitivity.

#### 2.2.3. Specificity Identification

The results are shown in Table 3. Among the DON pAbs prepared by the four methods, the CDI method exhibited the best sensitivity, with 100% recognition of DON. The I*C*_50_ was 47.75 ng/mL, and except for the slightly higher cross-reaction rate with 3-AcDON (35.3%), there was no cross-reaction with other compounds, indicating good specificity of the DON pAb. For the DON pAbs prepared by the other three methods, the sensitivity was poor, and the cross-reaction rate with 3-AcDON was also high. The results showed that the best artificial antigen synthetic method was the CDI method, which produced the highest titre, the best sensitivity, and the strongest specificity.

## 3. Discussion

### 3.1. Molecular Design of DON Haptens

DON has a special molecular structure, and the epoxide at the C12/13 position of DON is considered to be a key site of toxicity and to play a key role in inhibiting cellular protein synthesis [21,22,23]. Studies have shown that the opening of epoxides leads to the loss of DON toxicity [24]. In addition, there are hydroxyl groups (−OH) at the C3, C7, and C15 sites and carbonyl groups (−C=O−) at the C8 site, but these sites are not the active groups required for direct coupling with proteins, so appropriate active groups need to be introduced. Theoretically, these four sites can be introduced into the bridge structure; however, due to the influence of the carbonyl group on C8, the hydroxyl group on C7 is inactive, and it is generally difficult to generate derivative reactions. Therefore, in the work of most of the literature reporting the preparation of DON antibodies, DON hapten-derived sites were selected, and one of the hydroxyl groups was selectively used to derive the post-coupling protein. For example, Casale et al. [25] prepared an antigen by activating the C8 carbonyl group to immunise mice, and the antibody obtained was not specific. However, after immunising mice with an artificial antigen prepared by activating hydroxyl groups on C3, a hybridoma cell line stably secreting anti-DON mAbs was obtained, and an immunological detection method was established. Some studies, such as that of Li et al. [26], reported that good artificial antigens were also obtained by first blocking the hydroxyl groups at position 7 and position 15 and then using the hydroxyl group at position 3 for derivation. However, the preparation processes of these methods are relatively complex, and their operation is tedious. Considering the problems of sensitivity and specificity, four different methods were used for hapten derivation in this study. The first two methods were relatively simple and conducted without blocking any active site—namely, introducing the active group carbonyl (−C=O−) by the CDI method, and introducing the active group carboxyl (−COOH) by the maleic anhydride (MA) reaction to synthesise haptens. The latter two methods were more complicated. First, the hydroxyl groups (−OH) at position 7 and position 15 of DON were blocked with 1-butylboronic acid, and then the hydroxyl group (−OH) at position 3 was reacted with succinic anhydride (SA) and glutaric anhydride (GA) to introduce the active carboxyl group (−COOH) to synthesise haptens.

### 3.2. Preparation Method of DON Artificial Antigens

In this study, DON hapten was modified to introduce carbonyl groups and carboxyl groups and then coupled with proteins to prepare artificial antigens. Small-molecule haptens containing carbonyl and carboxyl groups can be coupled with the carrier protein by the CDI method, EDC method, IBCF method, NHS method, etc.; all of these are reactions in which the amino group (−NH_2_) in the carrier protein is coupled with the carbonyl (−C=O−) or carboxyl group (−COOH) in the hapten to form an amide bond (−CO−) [27]. Cyclohexylcarbodiimide (DCC), as a dehydrating agent, plays an important role in peptide synthesis, but it is readily crosslinked with water, which leads to test failure, and it is soluble in organic solvents. In this test, a coupling protein (BSA) that needs to be dissolved in aqueous solution is used, which makes it difficult to achieve anhydrous reaction conditions. Therefore, this test is not suitable for preparing artificial antigens by the DCC method. In this study, four coupling methods—the CDI method, EDC method, IBCF method, and NHS method—were compared to prepare artificial antigens and test antigens through comparative tests. The results showed that the CDI method described by Maragos et al. [28] and the improved method had the best effects, and the 3-hydroxy group on the DON molecule had a greater chance of reacting. Therefore, in this study, the hydroxyl group at the C3 position of the DON molecule was successfully covalently coupled to the amino group of the carrier protein BSA by the CDI activation method to obtain the artificial antigen DON-BSA without blocking the hydroxyl group at any site, and the imidazole group was directly introduced to achieve coupling with the carrier protein BSA [29]. The artificial antigen synthesised by this method was not very different from that produced by the preparation method reported by Lee et al. [30] in the previous literature. This method has a good effect for immunising mice, and the test scheme is simple and easy to operate, reducing the economic cost required by the test. The artificial antigen synthesised by the EDC method is derived directly from DON’s reaction with maleic anhydride and does not need to be sealed to obtain a better artificial antigen and detection antigen. We identified the artificial antigen by IR, UV, SDS–PAGE, and other methods, which preliminarily proved that the artificial antigen was successfully coupled. However, in the end, the mice were immunised with the artificial antigen, and then the titre of polyantisera was determined by ELISA. It was observed that the mice had higher titres and their polyantisera could be inhibited by DON molecules, proving that the artificial antigen was successfully synthesised. Therefore, in the process of artificial antigen coupling, we should attempt to implement a variety of methods to prepare and compare the results of immunising animals in order to screen the best method for preparing artificial antigens.

### 3.3. Analysis of the Immunological Characteristics of DON pAb

In this study, the immunological characteristics of DON pAb were analysed by determining the titre, sensitivity, and specificity. First, the titre of DON pAb was determined by indirect ELISA. The artificial antigens synthesised by the four methods were used to immunise the mice, and each artificial antigen was used to immunise one group of mice among a total of four groups. One mouse with the highest titre was selected from each group, and a total of four mice were selected. The titres of the polyantiserum reached 1: (1.6 × 10^3^). The results showed that the titres of the four artificial antigens all met the requirements but, according to the evaluation of immune effects, the order from high to low was DON-BSA (CDI), DON-BSA (EDC), DON-BSA (IBCF), and DON-BSA (NHS). Second, the sensitivity of DON pAb was determined by indirect competitive ELISA (icELISA). The pAbs of all four mice inhibited DON, and all of them met the criteria of a linear relationship. The I*C*_50_ value for mice in the DON-BSA (CDI) group was 47.75 ng/mL, and the regression equation was y = −37.056x + 112.23, R² = 0.9744; this inhibition effect was the best. The second-best group was the DON-BSA (EDC) group, while the sensitivity of the DON-BSA (IBCF) and DON-BSA (NHS) groups was poor. In addition, the specificity of DON pAb was identified by a cross-reaction test. The results showed that the DON pAbs prepared by the four methods could recognise DON with 100%, and the CDI method had the strongest specificity. Except for the slightly higher cross-reaction rate with 3-AcDON (35.3%), there was no cross-reaction with other compounds. Because 3-Ac-DON was closer to DON in spatial structure—while the other toxins (15-Ac-DON, T-2 toxin, zearalenone (ZEN), and aflatoxin B1 (AFB1)) differed greatly from DON in spatial structure, so their cross-reaction rate was very low—it showed strong specificity. Compared with 3-AcDON, the cross-reaction rates of the other three methods were higher, indicating that their specificity was not strong. In summary, the best artificial antigen synthesis method was the CDI method, which produced the highest titre, the best sensitivity, and the strongest specificity, followed by the EDC method.

## 4. Materials and Methods

### 4.1. Reagents and Materials

The standards of DON, ZEN, AFB1, 3-AcDON, 15-AcDON, and T-2 toxin were purchased from Sigma-Aldrich Co., Ltd. (Augsburg, Germany). CDI, MA, THF, DMF, and EDC were all Pierce products. Chicken ovalbumin (OVA, MW: 45,000), BSA (MW: 66,000), horseradish peroxidase (HRP), Freund′s incomplete adjuvant (FIA), and Freund′s complete adjuvant (FCA) were all Gibco products, were purchased from Yunke Reagent Consumables Station, Zhongyuan District, Zhengzhou City (Zhengzhou, China). Tetramethylbenzidine (TMB) was provided by Shanghai Wulian Chemical Plant (Shanghai, China). Sheep anti-mouse IgG antibody—namely, enzyme-labelled secondary antibody (GAMIGG-HRP)—was provided by Beijing Solebo Technology Co., Ltd. (Beijing, China).

MD25 dialysis bags (8000–14,000 D), 96-well microplates, 30% acrylamide, sodium dodecyl sulphate (SDS), tetramethylethylenediamine (TEMED), ammonium persulfate (AP), SDS–PAGE protein loading buffer (5×) and pre-dyed rainbow molecular weight standard protein marker were purchased from Huamei Biotechnology Company (Shanghai, China). Phosphate buffer solution (PBS), carbonate buffer solution (CBS), washing solution (PBST), blocking solution (SPBST), substrate chromogenic solution (TMB), termination solution (2 M H_2_SO_4_), and solutions required for SDS–PAGE were all homemade in our laboratory. Three times distilled water used for the tests was purified by our laboratory. Female BALB/c mice (6–8 weeks old) were provided by Beijing SPF Biotechnology Company and raised under strictly controlled conditions in our laboratory’s animal room.

DON standard solution: 5 mg of DON was accurately weighed, 5 mL of acetonitrile was added to dissolve it, and the mother solution with a concentration of 1 mg/mL of DON was prepared. The mother solution was diluted with PBS according to the concentration of DON required in the experiment.

### 4.2. Instruments and Equipment

A Varioskan Flash multifunctional microplate reader was purchased from BRUKER (Wiesbaden, Germany). A TENSOR 27 infrared spectrometer and HJ-3 digital display constant-temperature magnetic stirrer were provided by Thermo Fisher Scientific (Waltham, MA, USA). A BSA224S-CW electronic analytical balance was purchased from Beckman Coulter (Bria, CA, USA). A DU 800 UV–visible spectrophotometer was provided by Sartorius (Gottingen, Germany). The gel imaging system and analysis software were purchased from Syngene. An ultrapure water instrument was provided by Shanghai Danding International Trade Co., Ltd. A DH-360 electric thermostatic incubator was purchased from Shanghai Longyue Instrument Equipment Co., Ltd. (Shanghai, China). The immunogen emulsifying machine was made by our laboratory. An ND 2000 ultra-micro nucleic acid protein analyser was provided by Beijing Zhongxing Weiye Instrument Equipment Co., Ltd. An MTN-2800D nitrogen-blowing concentration device was provided by Millipore (Boston, MA, USA).

### 4.3. Test Methods

#### 4.3.1. Preparation of DON Artificial Antigens

According to the specific active site and active groups on the molecular structure of DON (Figure 6), DON was selected as the starting material for the molecular design of DON haptens and the preparation of artificial antigens. The main methods were CDI, EDC, IBCF, and NHS couplings. BSA was used as a carrier protein to prepare the artificial antigens, and OVA was used to prepare the detection antigens.

① N,N′-carbonyldiimidazole (CDI) method: The hapten was synthesised by using the 3-hydroxy group of DON to react with the imidazolyl group on the CDI molecule to introduce the carbonyl active group (−C=O−). Under the action of EDC, the introduced −C=O− was coupled with the −NH_2_ of the carrier protein BSA in the form of a monoamide bond (−CONH−) to synthesise the artificial antigen DON-BSA. For specific operation steps, refer to the CDI method described by Maragos et al. [28], with improvements. The detection antigen DON-OVA was prepared by the same method. The synthesis route is shown in Figure 7.

② 1-Ethyl-3-(3-dimethylaminopropyl)carbodiimide hydrochloride (EDC) method: The hapten (DON-HM) was synthesised by using the 3-hydroxy group of DON to react with the carboxyl group of MA to introduce the active group (−COOH). Under the action of EDC, the introduced −COOH was coupled with the −NH_2_ of the carrier protein BSA in the form of a monoamide bond (−CONH−) to synthesise the artificial antigen DON-BSA. For specific operation steps, refer to the EDC method described by Sun Zhen et al. [31], with improvements. The detection antigen DON-OVA was prepared by the same method. The synthesis route is shown in Figure 8.

③ Isobutyl chloroformate (IBCF) method: According to the molecular structure of DON, the hydroxyl groups (−OH) at the 7th and 15th positions of DON were first blocked with 1-butylboronic acid, and then the hydroxyl group at the 3rd position of DON was reacted with succinic anhydride (SA) to introduce the active group (−COOH) to form the semiester compound DON-HS. Under the action of IBCF, the artificial antigen DON-BSA was synthesised by coupling of the carrier protein BSA with NH_2_ in the form of a monoamide bond (−CONH−). The preparation process of this method was improved according to the IBCF method described by Casale et al. [25] and Li et al. [26]. The detection antigen DON-OVA was prepared by the same method. The synthesis route is shown in Figure 9.

④ N-hydroxysuccinimide (NHS) method: According to the molecular structure of DON, the hydroxyl groups (−OH) at the 7th and 15th positions of DON were first blocked with 1-butylboronic acid, and then the hydroxyl group at the 3rd position of DON was reacted with glutaric anhydride (GA) to introduce the active group (−COOH) to form the semiester compound DON-HG. Under the action of NHS and EDC, the artificial antigen DON-BSA was synthesised by coupling of the carrier protein BSA with NH_2_in the form of a monoamide bond (−CONH−). The preparation process of this method was improved according to the NHS method described by Deng Shunzhou et al. [32] and Zhang Yan et al. [33]. The detection antigen DON-OVA was prepared by the same method. The synthesis route is shown in Figure 10.

#### 4.3.2. Identification of the Physical and Chemical Properties of the Artificial Antigen DON-BSA

##### Determination of Protein Content

The standard BSA solution was used as a positive control, and PBS was used as a blank control. The artificial antigen DON-BSA and the detection antigen DON-OVA were diluted in PBS, the ultra-micro nucleic acid protein analyser was used for determination, and then the protein contents of both were determined at the wavelength of 280 nm.

##### UV Scanning Identification

The mother liquor of DON at a concentration of 1 mg/mL was diluted with PBS to a concentration of 100 μg/mL. The artificial antigen DON-BSA was diluted to 1 mg/mL in PBS solution, and the concentration of BSA was adjusted to 1 mg/mL. PBS solution was used for the blank control. The concentrations of BSA, DON, and DON-BSA proteins in the PBS solutions were determined using a DU 800 UV-visible spectrophotometer scanning in the wavelength range of 200–400 nm. The UV absorption peak was scanned and analysed to determine whether the artificial antigen DON-BSA was synthesised successfully.

##### IR Scanning Identification

The artificial antigen DON-BSA was freeze-dried in a freeze-dryer for two days, and then 1 mg was weighed and added to 200–250 mg of dried KBr, ground and mixed in an agate mortar, and irradiated with an infrared lamp. The time of pressing was 10 min, and the pressure was 10 MPa. The KBr transparent sample tablet was 1 mm thick [34]. KBr sample sheets of DON and BSA can be prepared by the same method.

##### SDS–PAGE Identification Method

A concentration gel and separation gel with concentrations of 5% and 12%, respectively, were used with voltages of 80 V and 40 V. The standard band was the pre-stained rainbow molecular weight standard protein marker, and the sample loading buffer was SDS–PAGE Sample Loading Buffer (5×), with 8 μg/well of protein loading and 10 μL/well of sample loading. At the same time, DON-BSA and BSA were subjected to SDS–PAGE electrophoresis and stained with Coomassie Brilliant Blue. The gel imaging system was used to capture photographs, and the test results were analysed and identified.

#### 4.3.3. Identification of Immunological Characteristics of Artificial Antigens

##### Preparation of DON pAb

Six-week-old female BALB/c mice were immunised with the artificial antigens DON-BSA (CDI), DON-BSA (EDC), DON-BSA (IBCF), and DON-BSA (NHS) synthesised by four different conjugation methods, and five mice were included in each of the four corresponding groups. The method of immunisation was a small dose and a long time interval. First, 1 mL of immunogen was prepared with sterile PBS, and the immune dose was 30 µg/mouse, which was injected subcutaneously into the back at 4 points. The number of immunisations was 5, except for the first immunisation with mixed emulsification of FCA, and the rest were all immunisations with FIA. The interval between the first immunisation and the second immunisation was 1 month, and the following intervals were 3 weeks. Ten days after the fifth immunisation (the last immunisation), the tail tip was cut off with small scissors for blood collection and centrifuged immediately, and DON polyantiserum was obtained. When blood was collected, the collected serum was diluted 1:100 with diluent PBS and centrifuged for 10 min at 3000 r/min. After centrifugation, the supernatant was stored in a refrigerator at 4 °C for later use. 

##### Determination of the Titre of DON pAb

The titre of DON pAb was determined by indirect ELISA. The effective positive titre was evaluated by calculating the A450 nm value of the wells to be tested as ≥2.1 times the A450 nm value of the negative wells. The operation procedure was carried out according to the conventional operation method.

##### Determination of DON pAb Sensitivity

The sensitivity of DON pAb was determined by icELISA. The working concentration of pAb was measured by the concentration of A450 nm ≈ 1.0 when the titre was measured. The experimental results were used to calculate the inhibition rate of anti-DON pAb, and the curve was generated and fitted. The correlation coefficient and regression equation were determined from the inhibition curve, and the I*C*_50_ of DON was calculated, which was then used as the result for measuring sensitivity. The operation procedure was carried out according to the conventional operation method.

##### Specific Identification of DON pAb

The specificity of the DON pAb was determined by a cross-reaction test. The half-maximal inhibitory concentration of pAb on functional analogues was determined by icELISA. The functional analogues included DON, 3-AcDON, 15-AcDON, T-2 toxin, ZEN, AFB1, etc. In addition, the cross-reaction rate (CR%) is the percentage of the half-maximal inhibitory concentration of pAb on DON divided by the half-maximal inhibitory concentration of pAb on other functional analogues. The formula is as follows:

CR% [35] = I*C*_50_ value of pAb versus DON/*IC*_50_ value of pAb versus other functional analogues × 100%

## 5. Conclusions

In this study, based on a large number of references and according to the particularity of DON’s molecular structure, artificial antigens were prepared by four coupling methods—the CDI method, EDC method, IBCF method, and NHS method—and the A280 nm protein concentrations of the four artificial antigens were determined using an ultra-micro nucleic acid protein analyser. Finally, the artificial antigens were identified by UV, IR, SDS–PAGE, and animal immunity. The results showed that the CDI method had the best effect among the four artificial antigen synthesis methods, and the DON pAb produced by animal immunity yielded the highest titre, the best sensitivity, and the strongest specificity. Therefore, this study provides a good antigen basis for the preparation of anti-DON mAbs in the next step.

## Figures and Tables

**Figure 1 molecules-28-02789-f001:**
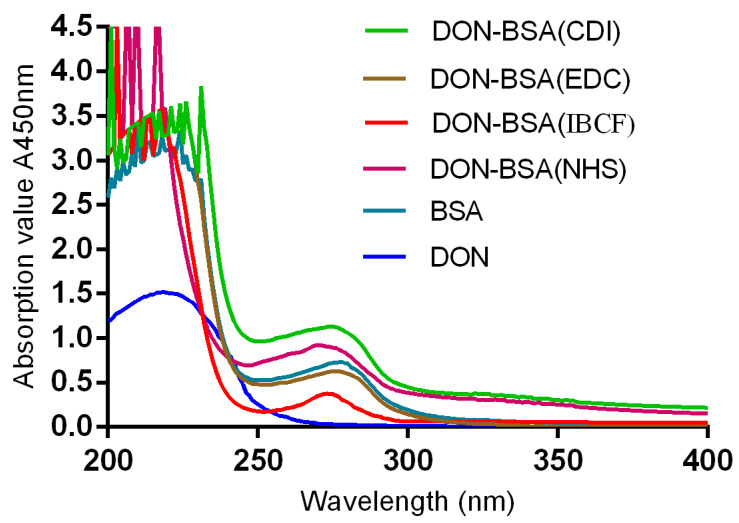
The UV identification of four artificial antigens (DON-BSA). UV: ultraviolet; DON-BSA: deoxynivalenol-bovine serum albumin; BSA: bovine serum albumin; CDI: N,N′-carbonyldiimidazole; EDC: 1-ethyl-3-(3-dimethylaminopropyl)carbodiimide hydrochloride; IBCF: isobutyl chloroformate; NHS: N-hydroxysuccinimide.

**Figure 2 molecules-28-02789-f002:**
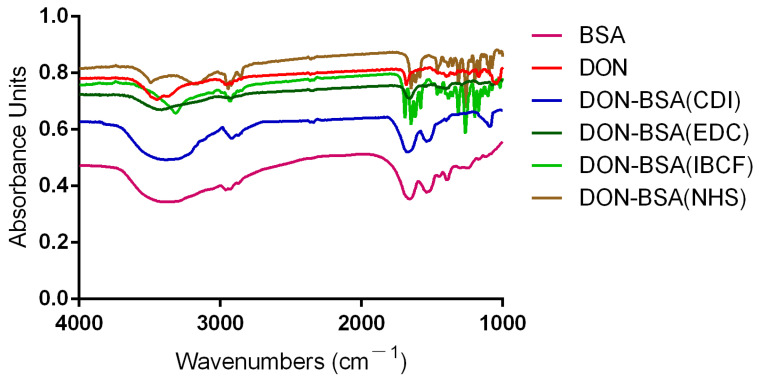
The IR identification of four artificial antigens (DON-BSA).

**Figure 3 molecules-28-02789-f003:**
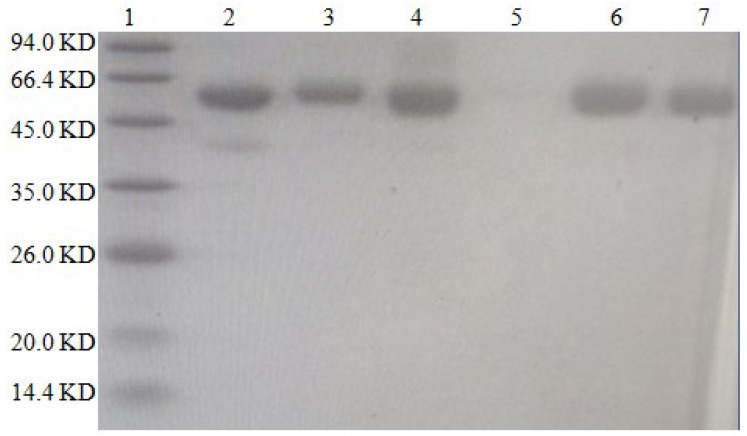
The SDS–PAGE characteristic of four artificial antigens for DON-BSA. 1—Maker; 2—BSA; 3—DON-BSA (CDI); 4—DON-BSA (EDC); 5—Blank; 6—DON-BSA (IBCF); 7—DON-BSA (NHS).

**Figure 4 molecules-28-02789-f004:**
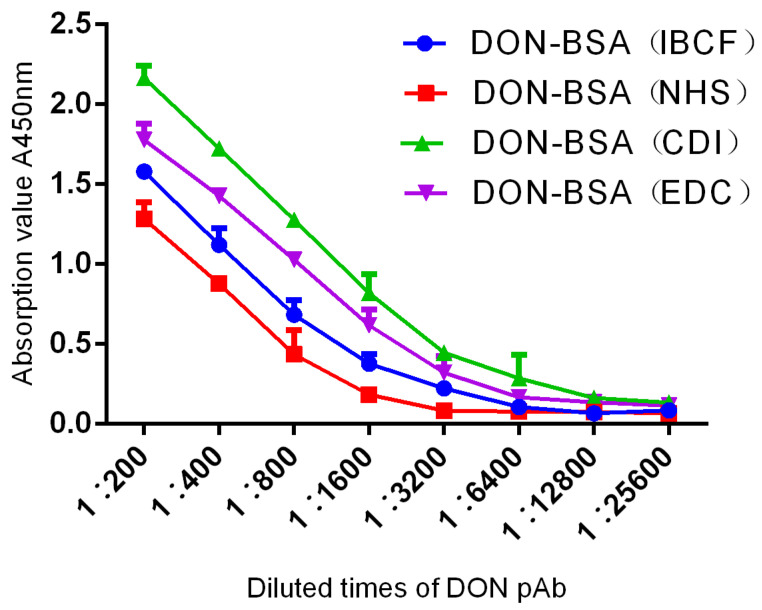
The titre curves of DON pAb identification.

**Figure 5 molecules-28-02789-f005:**
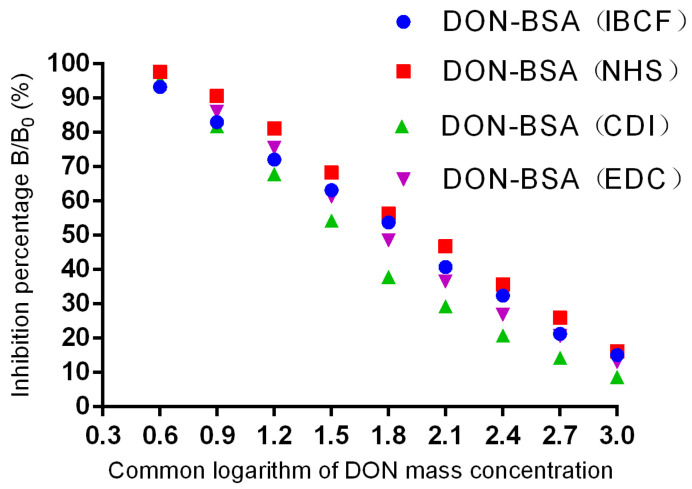
Inhibition curves of DON pAb.

**Figure 6 molecules-28-02789-f006:**
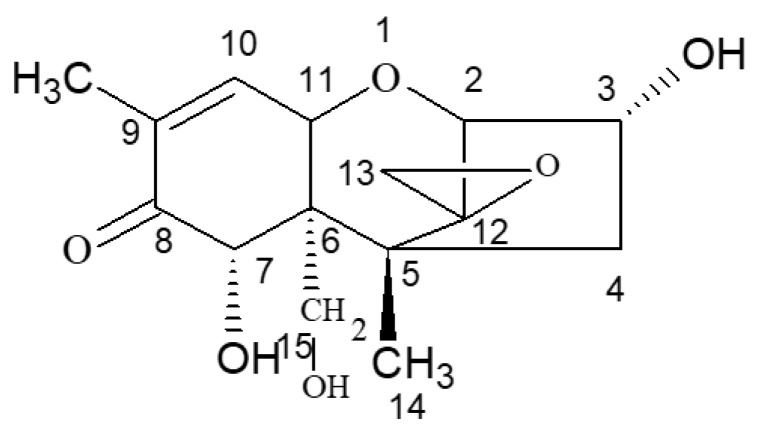
Molecular structure of DON.

**Figure 7 molecules-28-02789-f007:**
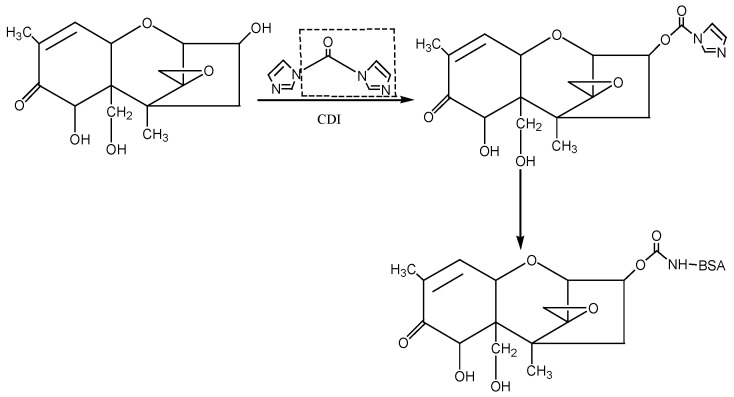
The synthesis route of DON-BSA by the CDI method.

**Figure 8 molecules-28-02789-f008:**
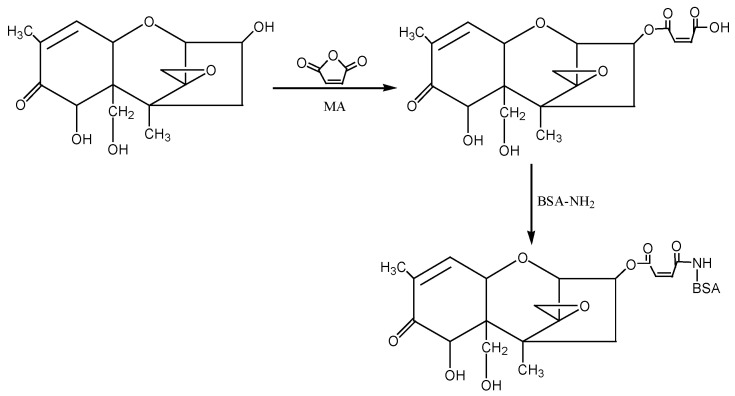
The synthesis route of DON-BSA by the EDC method.

**Figure 9 molecules-28-02789-f009:**
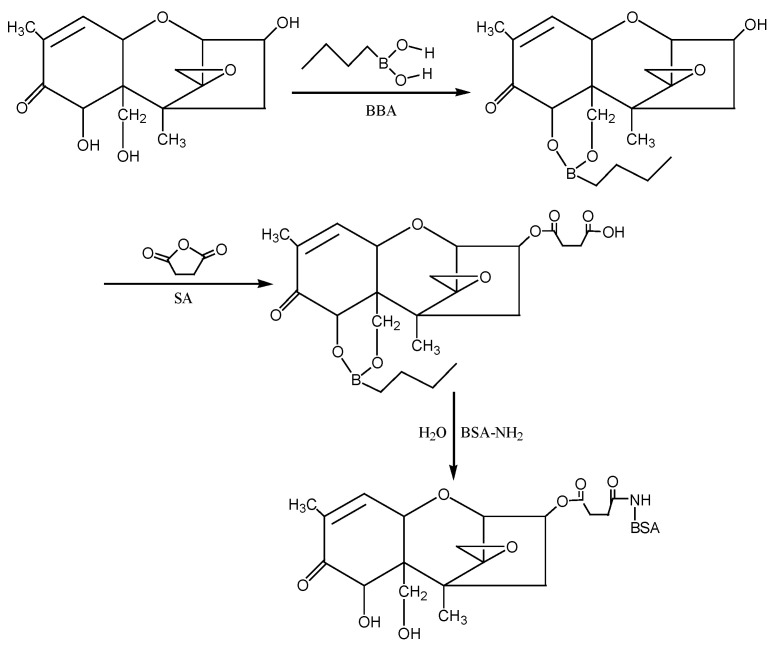
The synthesis route of DON-BSA by the IBCF method.

**Figure 10 molecules-28-02789-f010:**
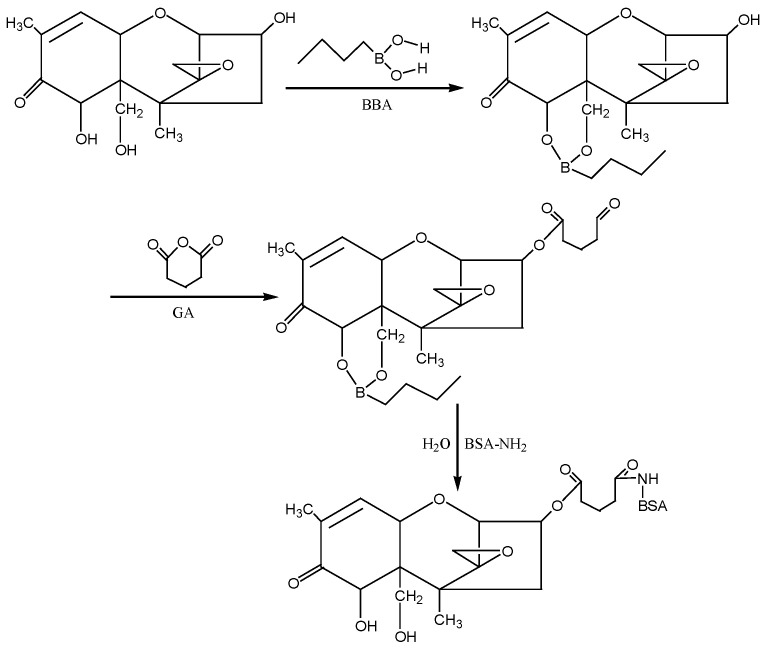
The synthesis route of DON-BSA by the NHS method.

**Table 1 molecules-28-02789-t001:** The titres determination of DON pAb of four mice.

Number	Diluted Multiple of Antiserum								Negative	Blank
	1:200	1:400	1:800	1:1600	1:3200	1:6400	1:12,800	1:25,600		
**EDC**	1.779	1.43	1.029	0.617	0.323	0.169	0.136	0.115	0.064	0.055
NHS	1.285	0.878	0.569	0.182	0.085	0.079	0.075	0.068	0.062	0.060
CDI	2.182	1.724	1.279	0.809	0.446	0.256	0.164	0.135	0.089	0.075
IBCF	1.579	1.123	0.789	0.445	0.223	0.107	0.097	0.086	0.085	0.072

Note: EDC: 1-ethyl-3-(3-dimethylaminopropyl)carbodiimide hydrochloride; NHS: N-hydroxysuccinimide; CDI: N,N′-carbonyldiimidazole; IBCF: isobutyl chloroformate.

**Table 2 molecules-28-02789-t002:** Regression equation, R², and I*C*_50_ for DON pAb.

Serial Number	Regression Equation	R^2^ Value	I*C*_50_ (ng/mL)
DON-BSA (EDC)	y = −36.744x + 117	0.98	66.53
DON-BSA (NHS)	y = −35.206x + 119.96	0.997	97.05
DON-BSA (CDI)	y = −37.056x + 112.23	0.97	47.75
DON-BSA (IBCF)	y = −33.717x + 113.29	0.997	75.34

**Table 3 molecules-28-02789-t003:** The specific identification of pAb.

Compounds	DON-BSA (CDI)		DON-BSA (EDC)		DON-BSA (IBCF)		DON-BSA (HS)	
	I*C*_50_ (ng/mL)	CR (%)	I*C*_50_ (ng/mL)	CR (%)	I*C*_50_ (ng/mL)	CR (%)	I*C*_50_ (ng/mL)	CR (%)
DON	47.75	100	70.96	100	91.35	100	112.72	100
3-AcDON	135.2	35.3	121.9	58.2	126.5	72.2	138.4	81.4
15-AcDON	>6.4 × 10^3^	<1.0	>6.4 × 10^3^	<1.0	>6.4 × 10^3^	<1.0	>6.4 × 10^3^	<1.0
T-2 toxin	>6.4 × 10^3^	<1.0	>6.4 × 10^3^	<1.0	>6.4 × 10^3^	<1.0	>6.4 × 10^3^	<1.0
ZEN	>6.4 × 10^3^	<1.0	>6.4 × 10^3^	<1.0	>6.4 × 10^3^	<1.0	>6.4 × 10^3^	<1.0
AFB1	>6.4 × 10^3^	<1.0	>6.4 × 10^3^	<1.0	>6.4 × 10^3^	<1.0	>6.4 × 10^3^	<1.0

Note: DON: deoxynivalenol; 3-AcDON: 3-acetyldeoxynivalenol; 15-AcDON: 15-acetyldeoxynivalenol; ZEN: zearalenone; AFB1: aflatoxin B1.

## Data Availability

Not applicable.

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
