# Peer review of "Comparison of Synthetic Methods and Identification of Several Artificial Antigens of Deoxynivalenol"

_molecules, 2023, doi:10.3390/molecules28062789_

Round 1

Reviewer 1 Report

This paper describes the proper several Comparison of Synthetic Methods and Identification Antigens of Deoxynivalenol. The paper could be published after major revisions. 

Figures 7,8,9 and 10 contain Chinese letters. It can not be acceptable.

All the reactions are provided without yields.

The confirmation of the intermediates (for each reaction) should be added. For example NMR spectra, can go to the sup. Material or the values for theoretical and experimental shifts. Otherwise the theoretical routes for the synthesis can be not admitted.

How was the IC50 calculated? This parameter can be calculated depending on the data fitting providing different values.

Table 2. mL instead of ml

 R2 should be 0.98, 0.997, 0.97 and 0.997. This parameter is depending on the number of nines. The rest of the numbers should be omitted.

Figure 5 should be presented with the linear fitting. It is not acceptable dots + lines.

Figure 2. No Y axis is presented. The figure must be presented with the attribution of the bands.

In general, the paper is hard to read. Many abbreviations are included. Please try to reduce them.

Minor errors:

Line 232. “This may be because 3-Ac-DON is more similar 231 to DON in spatial structure, while the other toxins 15-Ac-DON, T-2 toxin, ZEN, and AFB1 232 are significantly different from DON in spatial structure, resulting in a low cross-reaction 233 rate”. This sentence must be rewritten.

Reviewer 2 Report

This work is devoted to the study of four conjugates of deoxynivalenol (DON) with a carrier protein. The task of developing accurate and sensitive methods for detection DON is of immediate interest.

However, in my opinion, the scientific novelty of this work is not disclosed in the manuscript.

The fact is that various commercial monoclonal antibodies to DON, various polyclonal antibodies, and various kits for the detection of DON are currently available. The methods used by the authors to create conjugates are also standard. What is unique about this work?

In addition to this main disadvantage, there are the following issues:

In the introduction, the authors lead the reader to the idea of the need to create monoclonal antibodies against DON. At the same time, polyclonal antibodies are actually studied in the work, which seems somewhat contradictory.

I strongly recommend to present data on the study of conjugates obtained at different ratios of [DON] : [carrier protein] : [crosslinking agent].

I recommend to demonstrate an immunoassay for DON detection (based on the data obtained) and demonstrate its advantages over the existing one.

I recommend a side-by-side comparison of obtained pAb with commercially available antibodies, both polyclonal and monoclonal.

Since the work uses experiments with laboratory animals, it is necessary to indicate the number and date of the ethics committee protocol.

Some of the images have abbreviations not entered, as well as fragments not translated into English.

Round 2

Reviewer 1 Report

This minor revision must be done before publication. In figure 5 the authors say that a linear regression can be not achieved for all the graphs. My suggestion is that they have to eliminate the lines, leaving only the points in the graph.

Reviewer 2 Report

I got answers to most of my questions and believe that the article can be published in the present form.
